# FKBP38 Regulates Self-Renewal and Survival of GBM Neurospheres

**DOI:** 10.3390/cells12212562

**Published:** 2023-11-02

**Authors:** Aimee L. Dowling, Stuart Walbridge, Celine Ertekin, Sriya Namagiri, Krystal Camacho, Ashis Chowdhury, Jean-Paul Bryant, Eric Kohut, John D. Heiss, Desmond A. Brown, Sangamesh G. Kumbar, Yeshavanth Kumar Banasavadi-Siddegowda

**Affiliations:** 1Molecular & Therapeutics Unit, Surgical Neurology Branch, National Institute of Neurological Disorders and Stroke, National Institutes of Health, Bethesda, MD 20892, USAashis.chowdhury@nih.gov (A.C.); jeanpaulbryant@gmail.com (J.-P.B.); eric.kohut@nih.gov (E.K.); 2Clinical Neurology Unit, Surgical Neurology Branch, National Institute of Neurological Disorders and Stroke, National Institutes of Health, Bethesda, MD 20892, USA; 3Neurosurgical Oncology Unit, Surgical Neurology Branch, National Institute of Neurological Disorders and Stroke, National Institutes of Health, Bethesda, MD 20892, USA; desmond.brown@nih.gov; 4Department of Orthopedic Surgery, University of Connecticut Health, Farmington, CT 06030, USA; kumbar@uchc.edu

**Keywords:** FKBP38, apoptosis, autophagy, self-renewal, glioblastoma, PTEN, AKT

## Abstract

Glioblastoma is the most common malignant primary brain tumor. The outcome is dismal, despite the multimodal therapeutic approach that includes surgical resection, followed by radiation and chemotherapy. The quest for novel therapeutic targets to treat glioblastoma is underway. FKBP38, a member of the immunophilin family of proteins, is a multidomain protein that plays an important role in the regulation of cellular functions, including apoptosis and autophagy. In this study, we tested the role of FKBP38 in glioblastoma tumor biology. Expression of FKBP38 was upregulated in the patient-derived primary glioblastoma neurospheres (GBMNS), compared to normal human astrocytes. Attenuation of FKBP38 expression decreased the viability of GBMNSs and increased the caspase 3/7 activity, indicating that FKBP38 is required for the survival of GBMNSs. Further, the depletion of FKBP38 significantly reduced the number of neurospheres that were formed, implying that FKBP38 regulates the self-renewal of GBMNSs. Additionally, the transient knockdown of FKBP38 increased the LC3-II/I ratio, suggesting the induction of autophagy with the depletion of FKBP38. Further investigation showed that the negative regulation of autophagy by FKBP38 in GBMNSs is mediated through the JNK/C-Jun–PTEN–AKT pathway. In vivo, FKBP38 depletion significantly extended the survival of tumor-bearing mice. Overall, our results suggest that targeting FKBP38 imparts an anti-glioblastoma effect by inducing apoptosis and autophagy and thus can be a potential therapeutic target for glioblastoma therapy.

## 1. Introduction

Glioblastoma is the most common malignant primary brain tumor [1]. Despite maximal safe resection followed by chemoradiation and adjuvant chemotherapy, median survival is 15 months [1,2], and 5-year survival is only 6.8% [3]. Genetic heterogeneity, tumor-mediated immunosuppression, and the blood–brain barrier all result in impediments to successful therapy. Ultimately, the disease is universally fatal, and there is a dire need for novel therapeutic targets.

Immunophilins are a family of cis/trans peptidyl-prolyl isomerases (PPIases) that catalyze the interconversion between the cis and trans isomers of proline-containing peptide bonds [4]. These proteins serve as docking sites for immunosuppressive drugs, including FK506, rapamycin, and cyclosporin A, and are categorized as either cyclophilins (CYPs) or FK-506 binding proteins (FKBPs) [5]. While some FKBPs (such as the prototype FKBP12) have only the PPIase domain, other immunophilins contain additional domains, including tetratricopeptide repeat (TRP) and calmodulin-binding domains [5]. FKBP38 is the most unique immunophilin [6], as its PPIase activity is regulated by its calmodulin domain [7]. Furthermore, FKBP38 also contains an additional transmembrane domain that anchors it to the membranes of mitochondria and the endoplasmic reticulum [8,9]. FKBP38 regulates numerous pathways and functions, including cell size [10], development of neural tubes [11], mammalian target of rapamycin (mTOR) signaling [12], hypoxia response [13], viral replication [14], and protein folding and trafficking [15]. FKBP38 also exerts anti-apoptotic [8], pro-autophagic [16], and pro-mitophagic [17] effects, with broad potential implications for cancer pathogenesis. Consistently, FKBP38 has been implicated in numerous cancers [18,19]. We investigated the potential role of FKBP38 in glioblastoma pathogenesis. Expression of FKBP38 was elevated in patient-derived primary glioblastoma neurospheres, compared to normal human astrocytes. RNAi-mediated depletion of FKBP38 decreased viability by inducing apoptosis and autophagy in vitro and significantly increased median overall survival in mouse patient-derived xenografts. These findings provide evidence for FKBP38 as a potentially novel and effective therapeutic target in human glioblastoma.

## 2. Materials and Methods

### 2.1. Cell Culture

Glioblastoma neurosphere (GBMNS) cells were procured from Mayo Clinic (Rochester, MN) and cultured as described elsewhere [20]. Briefly, culture media was comprised of phenol red-free DMEM/F-12 (Invitrogen, Carlsbad, CA, USA), 1% penicillin–streptomycin, 2% B-27 without vitamin A (Invitrogen, Carlsbad, CA, USA), 1% sodium pyruvate, 20 ng/mL FGF, 20 ng/mL EGF, and 0.01% plasmocin. GBMNSs were cultured in low-attachment flasks in an incubator set to 37 °C and 5% CO_2_. Cultured cells were mycoplasma negative, as per the mycoplasma detection kit. Cells were authenticated by short tandem repeat profiling.

### 2.2. siRNA Transfection

FKBP38 target-specific (F38i) and scrambled-control siRNAs (Cntrl) were designed and synthesized (GE Dharmacon, Lafayette, CO, USA). Transfection of GBMNSs was performed using RNAi Max lipofectamine reagent (Invitrogen, Grand Island, NY, USA), per the manufacturer’s instructions.

### 2.3. Western Blot Analysis

Cell pellets were lysed using RIPA buffer (Sigma, St. Louis, MO, USA), including the protease and phosphatase inhibitor cocktail (Cell Signaling, Danvers, MA, USA), and protein concentration was quantified using a BCA assay (Bio-Rad, Irvine, CA, USA). Equal amounts of protein were loaded onto a NuPAGE 4–12% Bis-Tris gel (Invitrogen, Waltham, MA, USA) and transferred to an iBlot 2 NC membrane (Invitrogen, Waltham, MA, USA). The antibodies were used in varying dilutions from 1:1000–1:10,000; while the FKBP38 antibody was purchased from R&D Systems, Inc. (Minneapolis, MN, USA), GAPDH, PTEN, p-c-Jun, p-JNK, p-AKT, and LC3I/II antibodies were purchased from Cell Signaling (Danvers, MA, USA).

### 2.4. Antibody Array

The human apoptosis array and phospho-kinase antibody array (R&D Systems, Inc., Minneapolis, MN, USA) were performed on GBMNS cells transfected with Cntrl or F38i siRNAs, according to the manufacturer’s instructions. Briefly, cells were collected and lysed 72 h post-transfection, and lysates were incubated overnight at 4 °C with the membranes that were pre-coated with antibodies. Following the processing of incubated membranes as indicated in the kit, membranes were imaged by a Western blot imager.

### 2.5. Viability Assay

GBMNSs were seeded in a flat-bottom 96-well plate and transfected with Cntrl or F38i siRNAs. Cell viability was assessed via the XTT assay (ATCC, Manassas, VA, USA) per the manufacturer’s instructions. Briefly, XTT reagent was added to the experimental plate 72 h post-transfection. After 4 h of incubation, viability was assessed by the colorimetric method. Relative viability was determined by normalization to cells transfected with Cntrl siRNA.

### 2.6. Neurosphere Formation Assay 

GBMNSs were seeded in 96-well plates and transfected with Cntrl or F38i. The formation of neurospheres was observed daily over 5–7 days. The number of neurospheres formed in each well was counted and plotted against the number of cells seeded. Representative microscopic images were acquired.

### 2.7. Orthotopic Xenotransplantation of GBMNS

The animal studies were conducted per the principles and procedures outlined in the NIH Guide for the Care and Use of Animals and approved by the Animal Care and Use Committee of the National Institutes of Health (Protocol Number: ASP1503). Six-week-old athymic *Nu*/*Nu* mice were implanted intracranially with GBMNS 12 and GBMNS 43 (2 × 10^5^ cells per mouse), as described [20]. Ten mice were used per treatment group. Since the cells were pre-transfected with specific siRNAs, there was no room for randomization. Mice were anesthetized with isoflurane, ensuring the maintenance of core body temperature. Animals were then placed in a stereotactic frame. After meticulously cleaning the scalp with betadine and ethanol, a scalpel was used to make a small incision. A craniotomy drill equipped with a 1 mm bit was used to make an opening into the skull 2 mm lateral and 1 mm rostral to the bregma. GBMNSs (2 µL) pre-transfected with Cntrl or F38i siRNAs were implanted to a depth of 3 mm using a Hamilton syringe. Post-surgery, mice were weighed and observed until study endpoints were reached. Of note, mice losing more than 10% of their pre-implantation weight or those with signs of increased tumor burden, such as lethargy or a hunched back, were promptly euthanized to prevent suffering, as per study endpoint criteria.

### 2.8. Statistical Analysis

All experiments were performed in triplicates. Results are presented as mean ± standard deviation. Statistical analysis was carried out as described in the previous study [21]. The Student’s *t*-test and ANOVA were used to compare two or greater-than-two independent test groups, respectively. Survival analysis was performed using the log-rank and Kaplan–Meier tests. A *p*-value ≤ 0.05 was considered statistically significant.

## 3. Results

### 3.1. Expression of FKBP38 Is Higher in Patient-Derived Primary GBMNS

We assessed the expression level of FKBP38 in human GBMNSs and whether there was an increase in expression relative to normal human astrocytes (NHA). GBMNSs (GBMNS 12, GBMNS 43, GBMNS 28, and GBMNS 10) were probed for the expression of FKBP38 by Western blot analysis (Figure 1). There was a more than five-fold increase in the expression of FKBP38 in GBMNSs compared to normal human astrocytes (Figure 1B), suggesting that the expression of FKBP38 is upregulated in glioblastoma.

### 3.2. FKBP38 Knockdown Decreases the Viability and Self-Renewal Capacity of GBMNSs 

Initially, we probed the FKBP38 knockdown using four individual siRNAs in GBMNS 12 (Figure 2A). All four siRNAs were effective in depleting FKBP38. We chose F38i3 to test our hypothesis and reconfirmed FKBP38 knockdown in all the three GBMNSs used in this study (Figure 2B).

To determine whether FKBP38 is required for the survival of glioblastoma cells, we transfected GBMNSs with F38i3 and assessed viability at 72 h post-transfection using the XTT assay. Following FKBP38 knockdown, there was a statistically significant reduction in viability in all the GBMNSs that we tested (Figure 2C). These results suggest that FKBP38 depletion is deleterious for the survival of GBMNSs.

Self-renewal is one of the salient features of stem-like cells. As GBMNSs mainly constitute the stem-like glioblastoma cell population, we probed the role of FKBP38 in the self-renewal phenotype of GBMNSs (Figure 2D,E). With FKBP38 knockdown, the size (Figure 2D) and the number (Figure 2E) of the neurospheres were reduced significantly, suggesting that FKBP38 is required for both the survival and self-renewal of GBMNSs.

### 3.3. Depletion of FKBP38 Induces Apoptosis in GBMNS

FKBP38 inhibits apoptosis by recruiting the anti-apoptotic proteins Bcl-2 and Bcl-xl [8]. We assayed whether FKBP38 regulates apoptosis in GBMNSs. Caspase 3/7 activity was increased in FKBP38 knockdown cells, suggesting the induction of apoptosis (Figure 3A).

To determine the pathway through which FKBP38 regulates apoptosis in GBMNSs, we performed an antibody array for proteins involved in the apoptotic pathway. There was an increase in the expression of TRAIL R2, FADD, and cleaved caspase 3 (Figure 3B). To confirm this pathway, we assessed the expression of TRAIL R2, phospho-FADD, and cleaved caspase 3 using Western blot (Figure 3C). The phospho-FADD level increased in response to the FKBP38 knockdown. However, TRAIL R2 and cleaved caspase 3 were undetectable by Western blot. These results suggest that FKBP38 potentially regulates apoptosis in GBMNSs through the TRAIL R2-FADD axis.

### 3.4. FKBP38 Depletion Drives GBMNS toward Autophagy

FKBP38 is known to induce autophagy and mitophagy through the recruitment of LC3 to the mitochondrial membrane [16,17]. To investigate the role of FKBP38 in regulating autophagy in GBMNSs specifically, FKBP38 knockdown cells were probed for LC3 expression by Western blot. The depletion of FKBP38 increased the LC3-II/LC3-I ratio (Figure 4), suggesting that FKBP38 knockdown induces LC3-mediated autophagic activity in GBMNSs. This result directly counters the previously reported findings stating that the inhibition of FKBP38 abrogates autophagy in HEK293 and HeLa cells [17].

### 3.5. PI3K/AKT Mediates FKBP38-Regulated Autophagy in GBMNS

To explore the autophagic signaling pathway in the context of FKBP38, we performed an unbiased screening of phosphoproteins by antibody array (Figure 5A). FKBP38 depletion altered the phosphorylation profile of numerous proteins. Of particular interest was a reduction in the phosphorylation of c-Jun, implicating the JNK/c-Jun pathway regulation by FKBP38. Reduction in c-Jun phosphorylation in response to FKBP38 knockdown was confirmed by Western blot analysis (Figure 5B). As c-Jun regulates PTEN expression, we also probed for the expression of PTEN [22]. FKBP38 knockdown resulted in PTEN upregulation (Figure 5B). PTEN negatively regulates its downstream target, AKT [23,24]. Therefore, we also assayed for AKT activity. FKBP38 depletion induced PTEN expression and thus reduced AKT activity (Figure 5B).

Prior reports suggest that AKT activity negatively regulates autophagy [25]. Our results showed decreased AKT activity and increased autophagy with FKBP38 knockdown. To probe if AKT regulates autophagy in GBMNSs, we inhibited AKT activity in GBMNSs using perifosine and probed for LC3 expression. Similar to FKBP38 depletion, AKT inhibition increased the LC3-II/LC3-I ratio, confirming the negative regulation of autophagy by AKT in GBMNSs (Figure 5C). To further reconfirm that the FKBP38-AKT regulated autophagy, GBMNSs transfected with FKBP38 siRNA were treated with perifosine and probed for the expressions of phospho-AKT and LC3. Similar to Figure 4, FKBP38 depletion increased the LC3-II/LC3-I ratio. With perifosine treatment, irrespective of FKB38 status, the LC3-II/LC3-I ratio was upregulated, suggesting that blocking either the upstream target (FKBP38) or the intermediate target (PI3K/AKT) enhances the autophagic response in GBMNSs. Overall, our results provide evidence that FKBP38 depletion induces autophagy through the JNK/c-JUN–PTEN–AKT axis in human glioblastoma neurospheres.

### 3.6. FKBP38 Depletion Extends the Survival of Tumor-Bearing Mice

The potential clinical relevance of the in vitro findings was evaluated in a mouse orthotopic xenotransplantation model. We conducted FKBP38 knockdown in GBMNS 12 and GBMNS 43 by transfecting them with F38i3. At 72 h post-transfection, we performed intracranial implantation of FKBP38-intact and -depleted GBMNS 43 (Figure 6A) and GBMNS 12 (Figure 6B) in nude mice and followed their survival. FKBP38 knockdown resulted in a statistically significant improvement in overall survival in GBMNS 43 and GBMNS 12 tumor-bearing mice. FKBP38 knockdown extended the median survival of tumor-bearing mice by 7 days in both GBMNS 43 (23 to 30 days)- and GBMNS 12 (26 to 33 days)-implanted animals. These results provide strong evidence that FKBP38 knockdown has an anti-tumor effect in our in vivo patient-derived xenograft glioblastoma mouse models.

## 4. Discussion

The complex nature of glioblastoma tumor biology with profound cellular and genetic heterogeneity results in universal recurrence and uniform fatality [26,27,28,29]. In comparison to many non-CNS solid tumors for which a plethora of novel FDA-approved therapeutic options are available, there is a paucity of effective therapies for glioblastoma. While glioblastoma is comparatively rare, the high morbidity and mortality during peak productive years result in a tremendous economic burden, with public health ramifications. Thus, there is a dire need for novel therapeutic strategies.

In this study, we evaluated the role of FKBP38, a novel member of the immunophilin family, as an antineoplastic agent in glioblastoma. FKBP38 is a multidomain protein that regulates a variety of cellular functions, including cell size, apoptosis, development of neural tubes, mammalian target of rapamycin (mTOR) signaling, hypoxia response, viral replication, protein folding and trafficking, autophagy, and mitophagy [8,10,11,12,13,14,15,16]. Our results demonstrate that: (i) FKBP38 is upregulated in GBMNSs, compared to non-malignant human astrocytes (Figure 1); (ii) FKBP38 is required for the survival and self-renewal of GBMNSs (Figure 2); (iii) depletion of FKBP38 drives the cells toward apoptosis and autophagy (Figure 3 and Figure 4); (iv) FKBP38 inhibition-induced autophagy is regulated through the JNK/c-JUN–PTEN–AKT pathway (Figure 5); and (v) FKBP38 depletion extends the survival of tumor-bearing mice (Figure 6). The evidence put forth herein provide strong evidence for the potential utility of FKBP38 inhibition as a potentially effective therapeutic strategy in glioblastoma.

FKBP38 lacks the constitutive PPIase activity characteristic of the immunophilin family of proteins, despite possessing the PPIase domain [30]. In addition, FKBP38 does not bind FK506, nor does it bind CaN [8,31]. Perhaps as a result, FKBP38 has been underexplored, relative to several other members of the immunophilin family [19].

The role of FKBP38 as an apoptotic regulator has been called into question [32]. It has been suggested that FKBP38 actively translocates Bcl-2 from the ER to mitochondria, thus potentially preventing Bcl-2-mediated mitochondrial outer membrane permeabilization [8]. Additionally, FKBP38 binding to Bcl-2 prevents the latter from caspase-mediated degradation [18]. However, it has also been postulated that instead of anchoring Bcl-2 to the mitochondrial outer membrane, the more physiologically relevant role is related to the localization of the protein to the ER, where it acts as a co-chaperone in the folding and stabilization of proteins at the translational and post-translational levels [15]. Furthermore, the antineoplastic properties of FKBP38 have also been called into question, as the protein was shown to be downregulated in mouse models of metastatic cancers and shown to possess antineoplastic and anti-invasive functions when upregulated [33]. Thus, FKBP38 has been all but abandoned as an antineoplastic agent [32]. It is noteworthy that FKBP38 is predominantly expressed in the brain [30], and it has thus been postulated that perhaps the regulation of Bcl-2 by FKBP38 allows for the brain-specific regulation of apoptosis [32,34]. We found a consistent increase in Caspase 3/7 ratios across three distinct human glioblastoma neurosphere lines in response to FKBP38 knockdown, and this occurred in a mechanism dependent on FADD. This provides clear evidence for the role of FKBP38 in the regulation of apoptosis in human glioblastomas.

We also found evidence for the role of FKBP38 in the regulation of autophagy. Specifically, RNAi-mediated depletion of FKBP38 increased the LC3II/I ratio, consistent with an increase in autophagic pathways. Furthermore, this autophagic cascade was found to be the result of the activation of the JNK/c-JUN–PTEN–AKT axis in human glioblastoma neurospheres. Autophagy is a highly conserved catabolic mechanism that results in lysosomal degradation of misfolded proteins and altered organelles [35]. Cues for autophagic cascade initiation include nutrient starvation and hypoxia, with the end goal of homeostatic survival [36]. Consistently, autophagy has been shown to be upregulated in response to standard therapy, which has led some authors to consider whether autophagy induction imparts therapy resistance [37]. However, inadequate repairs and/or major or prolonged stress may put the cell in a state of extremis, resulting in autophagic death [38]. Autophagy may also exert direct antineoplastic effects [39]. In this study, we present strong in vitro and in vivo evidence for an overall antineoplastic effect of FKBP38 knockdown. Generally, apoptosis and autophagy are inversely correlated [40]; whenever there is a blockade of apoptotic response, non-apoptotic cell death pathways, such as autophagy, are typically activated. In normal cells, FKBP38 elicits an autophagic response through the recruitment of autophagy components, such as LC3A [17]. Therefore, we hypothesized that the induction of apoptosis secondary to FKBP38 depletion would be accompanied by a concomitant downregulation of autophagy. However, the opposite result was observed; we instead found a concomitant increase in both autophagy and apoptosis following FKBP38 RNAi-mediated downregulation. However, it is not yet apparent whether the induction of autophagy, in this context, antagonized apoptotic cell death (with the apoptotic event ultimately triumphant) or if it was synergistic with it. Nonetheless, this surprising and important finding suggests that FKBP38 depletion may potentially exert selective pressure against glioblastoma through the upregulation of both apoptotic and non-apoptotic (autophagy) cell death pathways. The context in which either pathway dominates remains unexplored, but our results clearly demonstrate that FKBP38 inhibition drives GBMNSs toward elimination, resulting in an antineoplastic effect, both in vitro and in vivo. Additional detailed studies will be required to dissect cell-, tissue-, and context-specific effects of FKBP38 on apoptosis and autophagy.

One clinically relevant and important inducer of autophagy is hypoxia, a low partial pressure of oxygen, which limits oxygen availability. Hypoxia is a central feature of many solid cancers and is critical in the pathogenic mechanisms in glioblastoma. Hypoxia induces neovascularization through the upregulation of hypoxia-inducible transcription factors (HIF) and vascular endothelial growth factors (VEGF). The resulting vessels are often small, abnormal, and frequently occluded, resulting in a selective pressure that favors the formation of new hypoxic areas and more aggressive cells—an unfortunate positive cascade [35]. In tissues with normal oxygen partial pressure, the HIF alpha subunit is rapidly degraded following hydroxylation by prolyl-4-hydroxylase 2 (PHD2) [13]. FKBP38 interacts with PHD2 and decreases its activity. Consistently, the RNAi-mediated depletion of FKBP38 has been shown to increase PHD2 activity, thus reducing HIF protein levels and the ability to withstand hypoxic conditions [41]. Interestingly, high levels of HIF activate the transcription factor BNIP3, which is implicated in the induction of autophagy [42]. It is likely that the FKBP38 induction of autophagy may be context-dependent. An in-depth comparative study exclusively focusing on the role of FKBP38 in the hypoxic and normoxic conditions in glioblastoma would decipher if targeting FKBP38 has a differential outcome, based on the normoxic and hypoxic statuses of the glioblastoma tumor cells.

## 5. Conclusions

Our study demonstrates that FKBP38 is upregulated in GBMNSs and is required for the survival and self-renewal of GBMNSs. Attenuation of FKBP38 drives GBMNSs toward apoptosis and autophagy, thus imparting anti-tumor efficacy in glioblastoma. In summary, FKBP38 is a potentially novel therapeutic target for glioblastoma that exerts its effects through the induction of both apoptosis and autophagy. Small molecule inhibitors and/or biological agents that inhibit and/or deplete FKBP38 may prove efficacious in glioblastoma, a uniformly fatal disease with limited therapeutic options.

## Figures and Tables

**Figure 1 cells-12-02562-f001:**
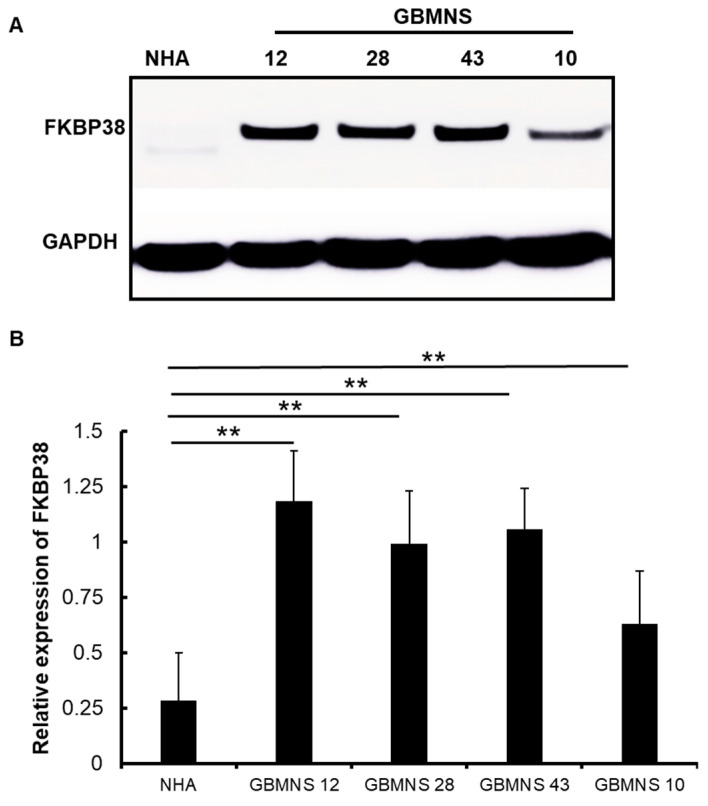
*Expression of FKBP38 is higher in patient-derived primary GBMNSs*: GBMNSs and normal human astrocytes (NHA) were probed for the expression of FKBP38. GAPDH was used as an internal control. (**A**) Western blot images showing the expressions of FKBP38 and GAPDH in NHA and GBMNSs 12, 28, 43, and 10. (**B**) The graph represents the quantification of the FKBP38 in GBMNSs and NHA from 3 independent biological samples (** *p* ≤ 0.01, n = 3).

**Figure 2 cells-12-02562-f002:**
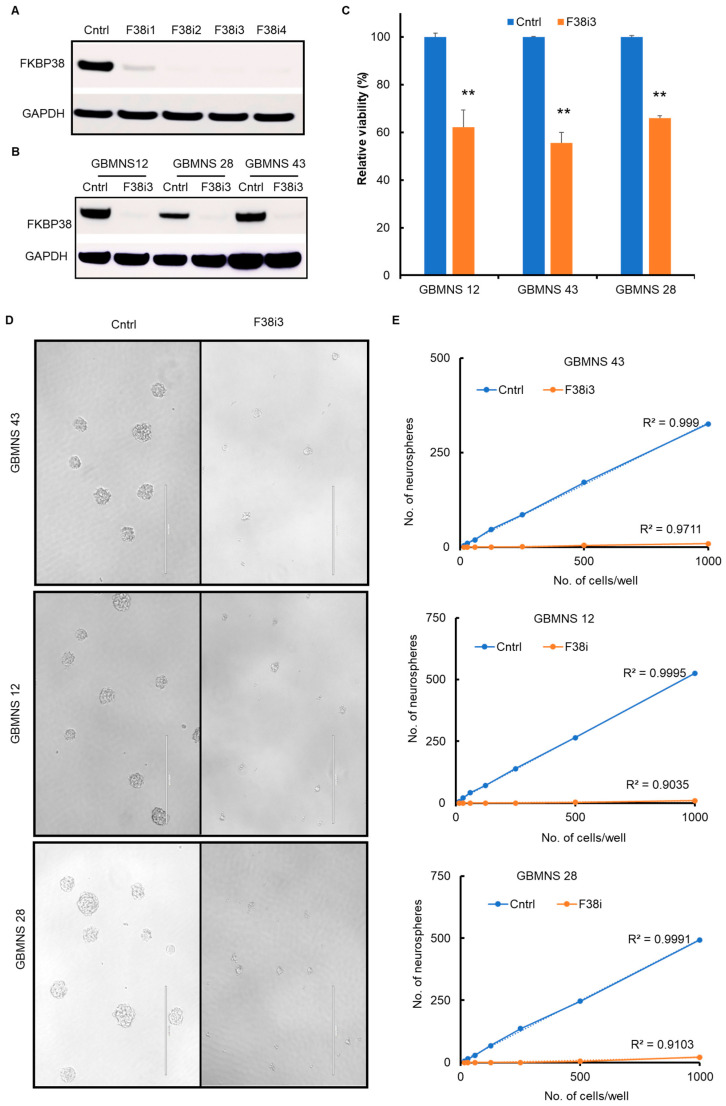
*FKBP38 knockdown decreases the viability of GBMNSs*: (**A**) GBMNS 12 transfected with 4 individual FKBP38 target-specific siRNAs (F38i1, F38i2, F38i3, and F38i4) or a scrambled siRNA (Cntrl) was probed for the expression of FKBP38. (**B**) GBMNSs were transfected with Cntrl or F38i3. At 72 h post-transfection, cells were collected and probed for FKBP38 by Western blot analysis. (**C**) GBMNSs transfected with F38i3 were subjected to a viability assay by the XTT assay. The graph represents the relative viability of Cntrl and F38i3 GBMNSs. (** *p* ≤ 0.01, n = 3). (**D**) Different numbers of cells ranging from 1000 to 15 cells were seeded in 96-well plates and transfected with F38i3. The formation of neurospheres was followed for 5–7 days. Representative images show the size and number of neurospheres. (**E**) The number of neurospheres formed was plotted against the number of cells seeded, as shown in the graph (n = 3).

**Figure 3 cells-12-02562-f003:**
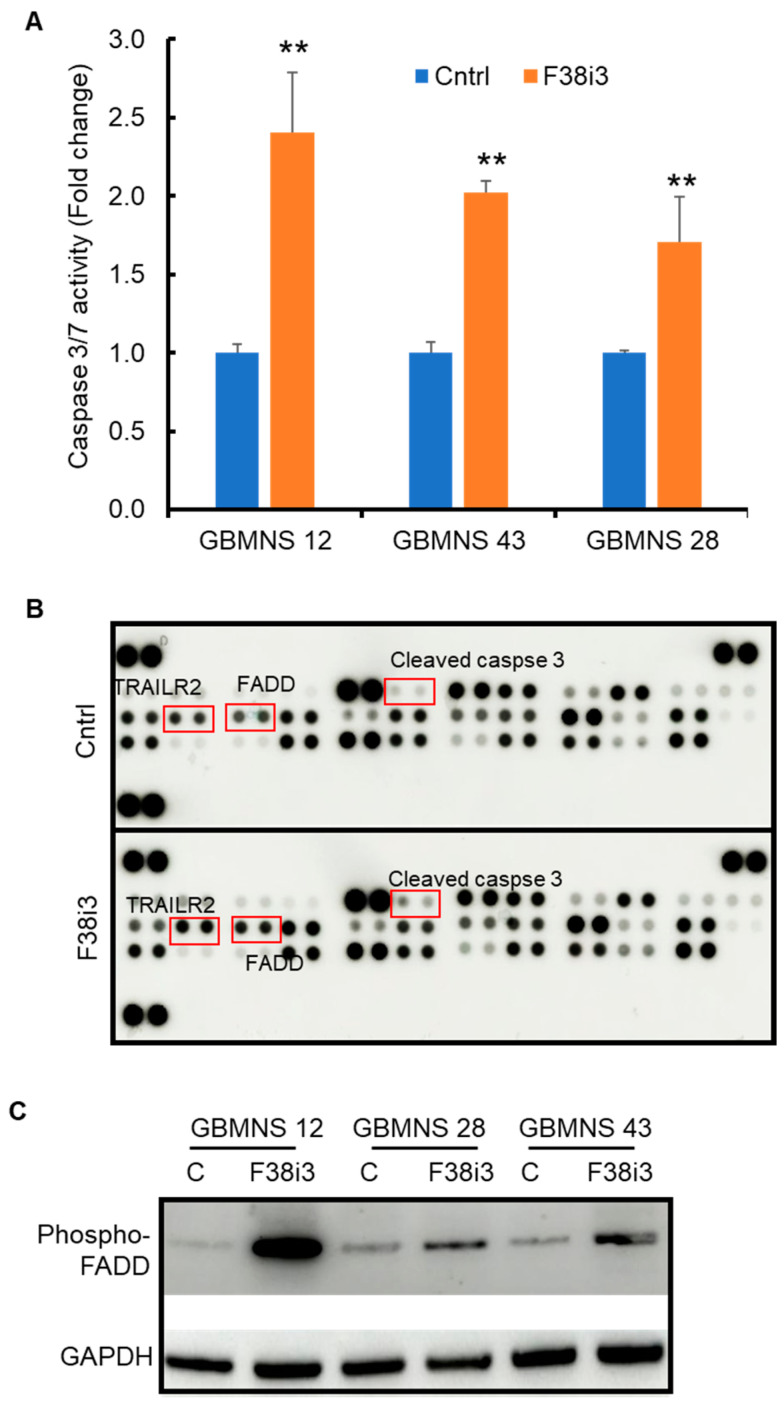
*Depletion of FKBP38 induces apoptosis in GBMNSs*: (**A**) GBMNSs transfected with F38i3 were probed for caspase 3/7 activity using Promega Caspase 3/7 activity assay. (** *p* ≤ 0.01, n = 3). (**B**) GBMNS 12 transfected with F38i3 was collected 72 h post-treatment and probed for apoptosis markers using an antibody array. (**C**) GBMNSs transfected with F38i3 were probed for phospho-FADD by Western blot.

**Figure 4 cells-12-02562-f004:**
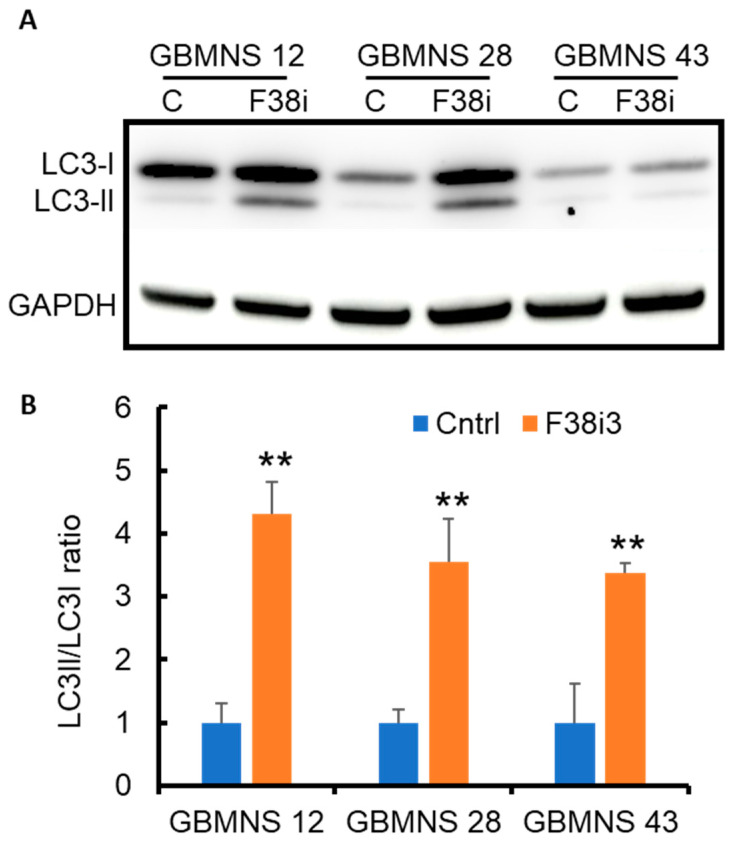
*FKBP38 depletion drives GBMNSs toward autophagy*: (**A**) GBMNSs were transfected with F38i3. At 72 h post-treatment, cells were collected and probed for LC3 expression by Western blot. (**B**) Quantification of the panel (**A**) showing the ratio between LC3-II and LC3-I. (** *p* ≤ 0.01, n = 3).

**Figure 5 cells-12-02562-f005:**
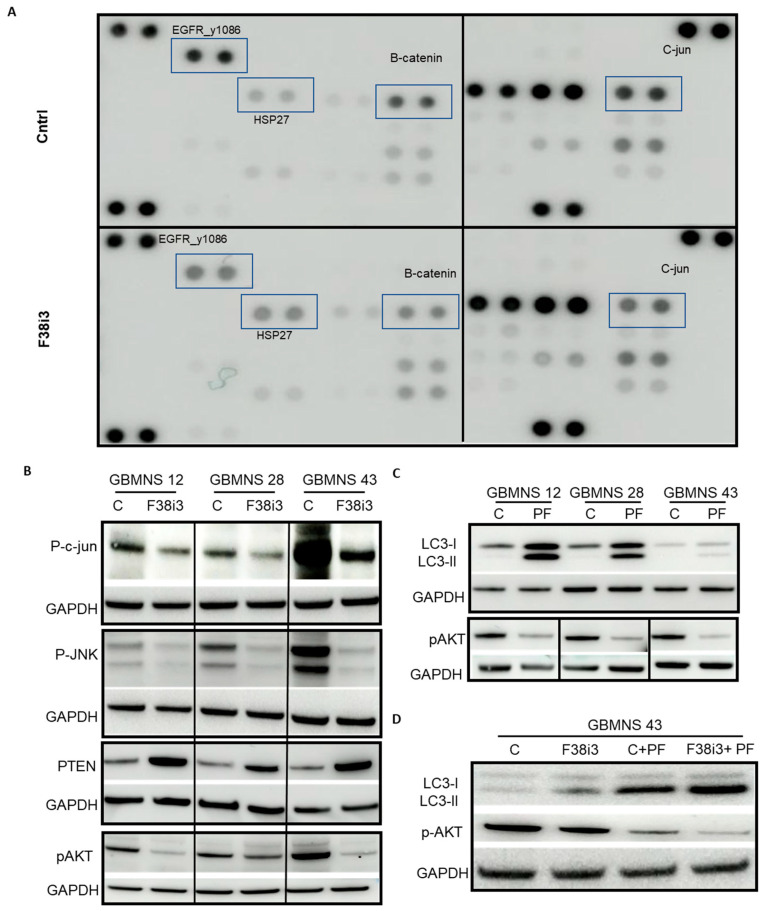
*PI3K/AKT mediates FKBP38-regulated autophagy in GBMNSs*: (**A**) GBMNS 12 was transfected with F38i3 and subjected to an antibody array. (**B**) GBMNSs transfected with F38i3 were probed for the indicated proteins. (**C**) GBMNSs treated with the AKT inhibitor perifosine (PF) were collected 24 h post-treatment and probed for phospho-AKT and LC3. (**D**) GBMNSs transfected with F38i3 were treated with AKT inhibitor perifosine (PF) and collected 24 h post-treatment to probe for phospho-AKT, LC3, and GAPDH.

**Figure 6 cells-12-02562-f006:**
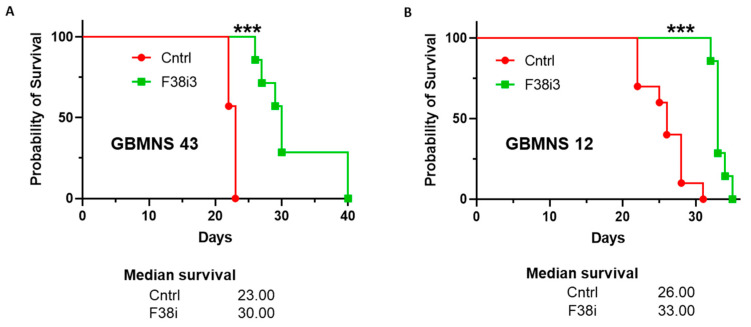
*FKBP38 depletion extends the survival of tumor-bearing mice*: FKBP38-intact (Cntrl) and -depleted (F38i3) GBMNS 43 (**A**) and GBMNS 12 (**B**) were implanted in nude mice, and their survival was followed. Shown are the Kaplan–Meier survival curves (*** *p* ≤ 0.01, n = 7).

## Data Availability

Not applicable.

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
