# Peer review of "FKBP38 Regulates Self-Renewal and Survival of GBM Neurospheres"

_cells, 2023, doi:10.3390/cells12212562_

Round 1
Reviewer 1 Report
Comments and Suggestions for Authors
In this study, the authors demonstrated that high expression of FKBP38 in GBM Neuro spheres. FKBP38 promote GBM Neuro spheres survival via Akt pathway. The writing and organize of this manuscript should be revised.
Comments:
In this manuscript, the word “probe” is unsuitable to use for western blot.
Figure 1B: The results should provide a standard deviation, not just from one result.
The expression of FKBP38 in GBM tumor cell lines should be provided.
Figure 2 provides too little information. Figure 2 should merge with Figure 4. The legend of Figure 2 did not have a title.
Figure 3B: The expressions of TRAILR2, FADD, and cleaved caspase-3 should be confirmed by western blot assay.
Figure 5B: The LC3-II should normalized to GAPDH, not LC3-I. The results should provide a standard deviation.
Figure 7. the authors should provide the tumor size and HE staining data.
Author Response
Reviewer 1:
1) In this study, the authors demonstrated that high expression of FKBP38 in GBM Neuro spheres. FKBP38 promote GBM Neuro spheres survival via Akt pathway. The writing and organize of this manuscript should be revised.
Response 1: We thank the reviewer for the comment. As per the reviewer’s suggestions we have written and reorganized the manuscript in this revised manuscript.
2) In this manuscript, the word “probe” is unsuitable to use for western blot.
Response 2: We thank the reviewer for the comment. We disagree with the reviewer as the word “probe” is relevant to explain Western blot.
3) Figure 1B: The results should provide a standard deviation, not just from one result. The expression of FKBP38 in GBM tumor cell lines should be provided.
Response 3: We thank the reviewer for the comment. We have included the standard deviation based on the quantification of 3 independent western blot images for the NHA and GBMNS.
4) Figure 2 provides too little information. Figure 2 should merge with Figure 4. The legend of Figure 2 did not have a title.
Response 4: We agree with the reviewer. As per the reviewer’s suggestion, we have merged Figure 4 with Figure 2. We have included the title in the revised manuscript.
5) Figure 3B: The expressions of TRAILR2, FADD, and cleaved caspase-3 should be confirmed by western blot assay.
Response 5: We were able to confirm the phospho-FADD expression by western blot.TRAILR2 and caspase3 expression were undetectable by western blot. For the confirmation of the caspase activity, we have furnished the caspase 3/7 activity as shown in Figure 3A.
6) Figure 5B: The LC3-II should normalized to GAPDH, not LC3-I. The results should provide a standard deviation.
Response 6: We thank the reviewer for the comment. In the revised manuscript, this will be Figure 4. First step: We normalized the LC3-II and LC3-I bands with the GAPDH. Then we calculated the ratio of LC3II/LC3I as it is the correct way to show the autophagy activity as per the previous publications (FKBP8 recruits LC3A to mediate Parkin‐independent mitophagy | EMBO reports (embopress.org). As per the reviewer’s suggestion, we have quantified the membrane from 3 independent experiments and plotted the graph with standard deviation.
7) Figure 7. The authors should provide the tumor size and HE staining data.
Response 7: We thank the reviewer for the comments. In the revised manuscript it will be Figure 6. We agree with the reviewer that H&E staining would provide tumor size information. To get this data we have to resubmit the amendment for our animal study protocol that might or might not be approved by the Animal Care and Use Committee of the National Institute of Health as this is complementary information. Implementation of animal study (if approved), collection of tumor samples, and staining would take additional time. Overall it might take 5-6 months to get this data. Hence we cannot do this experiment at this time.
Reviewer 2 Report
Comments and Suggestions for Authors
The manuscript “FKBP38 Regulates Self-Renewal and Survival of Gbm Neurospheres” by Aimee Dowling and eleven more colleagues investigated the potential role of FKBP38 in glioblastoma pathogenesis. Concludes that the xpression of FKBP38 was elevated in patient derived primary glioblastoma neurospheres compared to normal human astrocytes. Also, the RNAi-mediated depletion of FKBP38 decreased viability by inducing apoptosis and autophagy in vitro and significantly increased overall survival in mouse patient-derived xenografts. They suggest that FKBP38 is a potentially novel and effective therapeutic target in human glioblastoma.
The research is very straight forward and it is valid as an introduction to possible treatment options for glioblastoma. It is of potential interest to researchers and clinicians.
If there is not known previosly, the article and interest would be enhanced by immunohistochemistry on a human GBM showing expression in cells and cell types in the entire tumor. If it is known, please, include it in the Introduction section.
Regarding the edition:
Please, include titles on all figures. They improve the speed when viewing the study.
Figure 8 is confusing. Except for the arrows, it is identical. The effect on AP-1 promoter is not clear, the JNK/c-Jun complex is identical in both cases: what is the effect on AP-1 that causes PTEN to rise in one case and in the other to fall? This outline should be clearer and summarize the conclusions of the study.
If you know of any drug or FKBP38 modulator, you should explain it or comment on the possible use of F38i's.Also, Also, what final percentage would be modulated on a pathway so common (JNK/c-jun) for other molecular modulators.
Given the little bibliography on the topic, in the discussion they should contrast their results with those of Pistollato and al, (DOI: 10.1371/journal.pone.0006206).
Author Response
Reviewer 2:
1) The manuscript “FKBP38 Regulates Self-Renewal and Survival of Gbm Neurospheres” by Aimee Dowling and eleven more colleagues investigated the potential role of FKBP38 in glioblastoma pathogenesis. Concludes that the expression of FKBP38 was elevated in patient-derived primary glioblastoma neurospheres compared to normal human astrocytes. Also, the RNAi-mediated depletion of FKBP38 decreased viability by inducing apoptosis and autophagy in vitro and significantly increased overall survival in mouse patient-derived xenografts. They suggest that FKBP38 is a potentially novel and effective therapeutic target in human glioblastoma. The research is very straightforward forward and it is valid as an introduction to possible treatment options for glioblastoma. It is of potential interest to researchers and clinicians.
Response 1: We thank the reviewer for the comment.
2) If it is not known previously, the article and interest would be enhanced by immunohistochemistry on a human GBM showing expression in cells and cell types in the entire tumor. If it is known, please, include it in the Introduction section.
Response 2: We thank the reviewer for the comments. We did conduct IHC on the human GBM samples. Unfortunately, these samples did not stain for FKBP38. This is potentially because the FKBP38 antibodies that we tested failed to detect FKBP38 in the human GBM tumor samples.
3) Please, include titles on all figures. They improve the speed when viewing the study.
Response 3: We agree with the reviewer. In the revised manuscript we have included the titles for all the figures.
4) Figure 8 is confusing. Except for the arrows, it is identical. The effect on AP-1 promoter is not clear, the JNK/c-Jun complex is identical in both cases: what is the effect on AP-1 that causes PTEN to rise in one case and in the other to fall? This outline should be clearer and summarize the conclusions of the study.
Response 4: We thank the reviewer for the comment. We agree with the reviewer. As we did not study the effect of AP-1 on PTEN, to avoid confusion, we have removed the schematic representation (Figure 8) from the manuscript.
5) If you know of any drug or FKBP38 modulator, you should explain it or comment on the possible use of F38i's. Also, Also, what final percentage would be modulated on a pathway so common (JNK/c-jun) for other molecular modulators.
Response 5: We thank the reviewer for the comment. There is no known drug that specifically targets FKBP38. Developing a small molecule inhibitor for FKBP38 in collaboration with the Drug development core facility is one of the future directions for this project.
6) Given the little bibliography on the topic, in the discussion they should contrast their results with those of Pistollato and al, (DOI: 10.1371/journal.pone.0006206).
Response 6: We thank the reviewer for the comments. As suggested by the reviewer, we have revised the discussion comparing the results of pistollato et al.
Round 2
Reviewer 1 Report
Comments and Suggestions for Authors
The manuscript is well-revised according to the reviewer's suggestion.
Author Response
Thank you
Reviewer 2 Report
Comments and Suggestions for Authors
Most of the points in which I suggested improvements have been made. However, I continue to believe that if the aim of this article is to regulate FKBP38 in GBM, its demonstration in that tumor should be mandatory or, at least, state that an attempt was made to show it and it was not achieved.
Author Response
Thank you